# Brief communication: On calculating the sea-level contribution in marine ice-sheet models

Heiko Goelzer[1,2], Violaine Coulon[2], Frank Pattyn[2], Bas de Boer[3], Roderik van de Wal[1,4]

[1] Institute for Marine and Atmospheric research Utrecht, Utrecht University, Utrecht, the Netherlands
5 [2] Laboratoire de Glaciologie, Université Libre de Bruxelles, Brussels, Belgium
[3] Earth and Climate Cluster, Faculty of Science, Vrije Universiteit Amsterdam, the Netherlands
[4] Geosciences, Physical Geography, Utrecht University, Utrecht, the Netherlands

*Correspondence to*: Heiko Goelzer (h.goelzer@uu.nl)

10 **Abstract.**

Estimating the contribution of marine ice sheets to sea-level rise is complicated by ice grounded below sea level that is replaced by ocean water when melted. The common approach is to only consider the ice volume above flotation, defined as the volume of ice to be removed from an ice column to become afloat. With isostatic adjustment of the bedrock and external sea-level forcing that is not a result of mass changes of the ice sheet under consideration, this approach breaks down, because 15 ice volume above flotation can be modified without actual changes of the sea-level contribution. We discuss a consistent and generalised approach for estimating the sea-level contribution from marine ice sheets.

## 1. Introduction

Model simulations of past and future ice-sheet evolution are an important tool to understand and estimate the contribution of ice sheets to sea-level at different time scales (e.g. de Boer et al., 2015; Nowicki et al., 2016). The mass balance of ice sheets 20 is controlled by mass gain and loss at the upper, lower and lateral boundaries by melting or sublimation, by accumulation and freeze-on, and discharge of ice into the surrounding oceans. The sea-level contribution from an ice sheet can in principle be estimated through these different mass balance terms, but is in practice typically based on changes in one prognostic variable, *ice thickness*, and considering corrections for the ice grounded below sea level (e.g. Bamber et al., 2013). However, complications arise, especially for longer timescales, when isostatic adjustment of the bedrock is considered. The discussions 25 in this communication apply for ice-sheet models that include some form of a glacio-isostatic adjustment (GIA), but that are not coupled to the sea-level equation. While other examples exists (e.g. Gomez et al., 2013; de Boer et al., 2014), the models considered here typically account strictly for uplift or sinking of the bedrock beneath or proximal to an ice sheet, but do not include other (global) effects, such as sea-level changes due to changes in Earth's rotation and regional sea-level change due to changes in the Earth's gravitational field. However, the effect of mass changes from other ice sheets may be included in a 30 simplified form using an external sea-level forcing. Such forcing is decoupled from mass changes of the ice sheet itself and

prescribes sea-level changes in the model domain with the aim to capture its effect on ice flotation. The aim of this paper is to propose an approach to accurately estimate the contribution of the ice sheet in such a model to global-mean geocentric sea-level rise (see Gregory et al., 2019).

In our own ice-sheet modelling experience and from exchange with colleagues in different groups it is not always clear how the sea-level contribution should exactly be calculated and what corrections need to be applied. This goes hand in hand with a lack of documentation and transparency in the published literature on how the sea-level contribution is estimated in different models. With this brief communication, we hope to stimulate awareness and discussion in the community to improve on this situation. We caution that it is well possible that the proposed solutions or equivalent approaches are already in use in several models, since the fundamental ideas have already been laid out (e.g. Bamber et al., 2013; de Boer et al., 2015) and are straightforward to implement. Our aim here is to provide concrete guidelines and a central reference of best practices for ice-sheet modellers.

We describe in the following how to calculate the sea-level contribution for a situation without bedrock changes (Sec. 1), the effect of bedrock changes and how to account for them (Sec. 2 and 3), a density correction (Sec. 4) and modifications required when the model is forced by external sea-level changes (Sec. 5). We conclude with a realistic modelling example (Sec. 6) and a discussion (Sec. 7).

## 1. Estimating the sea-level contribution

If changes in the bedrock elevation due to isostatic adjustment are zero or very small, e.g. for centennial time scale simulations (e.g. Nowicki et al., 2016), the sea-level contribution of an ice sheet is typically computed from changes in total ice volume above flotation

$$V_{af} = \sum_n \left( H_n + min(b_n, 0) \frac{\rho_{ocean}}{\rho_{ice}} \right) \frac{1}{k_n^2} A_n, \tag{1}$$

where H is ice thickness, b is bedrock elevation (negative if below sea level) and e.g. $\rho_{ice}$=910 kg m$^{-3}$ and $\rho_{ocean}$ =1028 kg m$^{-3}$ are the densities of ice and ocean water, respectively. The sum is over the number $n$ of grid cells (elements) of an (un-) structured grid with area $A_n$. The unitless map scale factor k is applied when the model grid is laid out on a projected horizontal coordinate system, which is often the case for polar ice-sheet models (Snyder, 1987, Reerink et al. 2016). Below, we will often simplify the discussion in order to examine the interplay between ice sheet thickness, bedrock elevation, and sea level for a single column, which can be conceptualized as the values occurring in any single model grid cell or element (in map view). In that framework, we will refer to the limit ice thickness required for the ice to start floating as the *floatation thickness,* which is determined by the local bedrock elevation and sea level. $V_{af}$ of a column of ice grounded below sea-level may be interpreted as the amount of ice volume that has to be removed to reach the floatation thickness and for the column to start to float. This considers that floating ice is in hydrostatic balance with the surrounding water, and assumes that the ice does not contribute to sea-level changes when melted. In reality, however, densities of sea water and melted land ice

(freshwater) differ slightly, which is often neglected. An associated density correction is discussed below (Sec. 4). For ice grounded on land above sea-level, $b > 0$ and $V_{af} = H\,A_n * \frac{1}{k^2}$.

To estimate the ice volume in global sea-level equivalent (SLE$_{af}$ [m]), the total V$_{af}$ has to be converted into the volume it will occupy when added to the ocean assuming a sea-water density $\rho_{ocean}$=1028 kg m$^{-3}$ and divided by the ocean area A$_{ocean}$ of

typically 3.625 x 10$^{14}$ m$^2$ (Gregory et al., 2019).

$$SLE_{af} = \frac{V_{af}}{A_{ocean}}\frac{\rho_{ice}}{\rho_{ocean}}. \tag{2}$$

A$_{ocean}$ is assumed to be constant here, but on longer time scales this is not necessarily correct. Estimating changes in A$_{ocean}$ correctly would require a fully-coupled global ice sheet-GIA-sea-level model (e.g. Gomez et al., 2013; de Boer et al., 2014). The actual sea-level contribution of the modelled ice sheet (SLC) is typically calculated relative to a reference value, often the present day (modelled) configuration or the configuration at the start or end of an experiment.

$$SLC_{af} = -(SLE_{af} - SLE_{af}^{ref}). \tag{3}$$

Note that the minus sign in front of the parentheses in Eq. 3 is necessary since $SLE_{af}$ is a function of $V_{af}$, for which an increase over time is associated with a drop in sea level.

Depending on the amount of ice grounded below sea level, estimating the sea-level contribution instead from the entire grounded ice volume $V_{gr}$ (Eq. (4)) can lead to considerable biases and is only shown for comparison here.

$$SLC_{gr} = -\left[\frac{V_{gr}}{A_{ocean}}\frac{\rho_{ice}}{\rho_{ocean}} - \left(\frac{V_{gr}}{A_{ocean}}\frac{\rho_{ice}}{\rho_{ocean}}\right)^{ref}\right] \tag{4}$$

**2.  Effect of bedrock changes**

In this section we discuss additional considerations that are required when the model includes a GIA component that simulates bedrock changes. When changes in bedrock elevation occur under the ice, V$_{af}$ cannot always be used without a correction as basis for sea-level calculations, because isostatic uplift or lowering can modify V$_{af}$ without actual sea-level contribution. Figure 1 illustrates this problem for a single ice column with an uplift of the bedrock elevation (left to right in

each panel), where the bars indicate the bedrock and ice for different possible configurations. In case A, bedrock is already above sea level (i.e. V$_{af}$ includes all ice) and the vertical upward displacement has no apparent influence on the grounded configuration. In case B, ice is displaced upwards with the bedrock, the floatation thickness decreases and some of the ice is 'transformed' into ice above flotation. In case C a transition from floating to grounded ice occurs and in case D, ocean water is displaced by the rising bedrock.

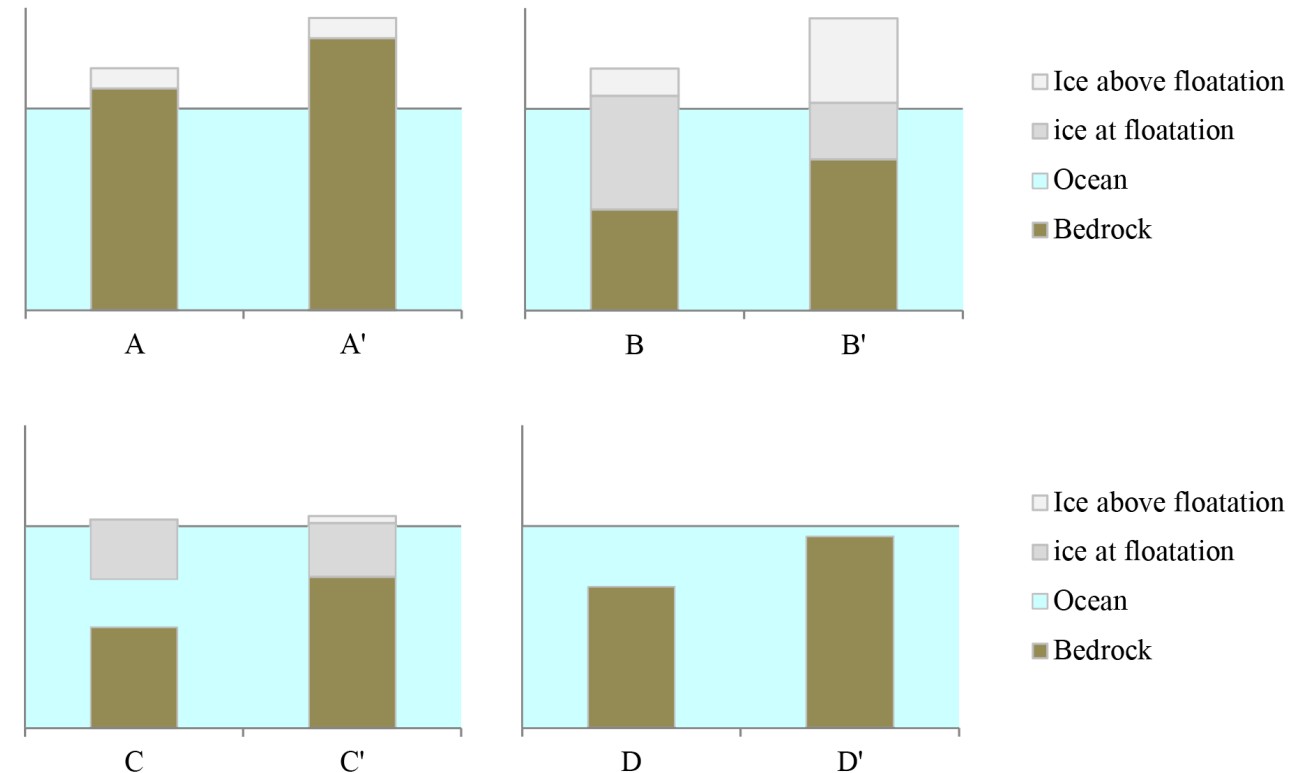

**Figure 1 Effect of bedrock changes. Different geometric configurations of ice, ocean and bedrock before and after (') a rise in bedrock elevation.**

The problem how to interpret these changes in sea-level contribution in the presence of bedrock changes is further illustrated by an evolution of one grid box in time (Figure 2a). If we compare between $t_1$ and $t_4$ and only look at the ice column, we could assume that there was no net sea-level contribution since the ice is just starting to float ($t_1$) or floating ($t_4$) in both cases. However, following the evolution through $t_2$ and $t_3$ gives rise to another interpretation. At $t_1$ the ice is just starting to float with a low bedrock elevation. The bedrock then rises ($t_2$) and subsequently ice is lost e.g. by surface melting ($t_3$). Finally, more ice is lost e.g. by basal melting and the ice is floating at $t_4$. From $t_1$ to $t_2$, ice is merely displaced by the bedrock, but the actual sea-level contribution occurs between $t_2$ and $t_3$ and equals the ice above flotation in $t_2$ and (by construction) also the bedrock displacement between $t_1$ and $t_4$.

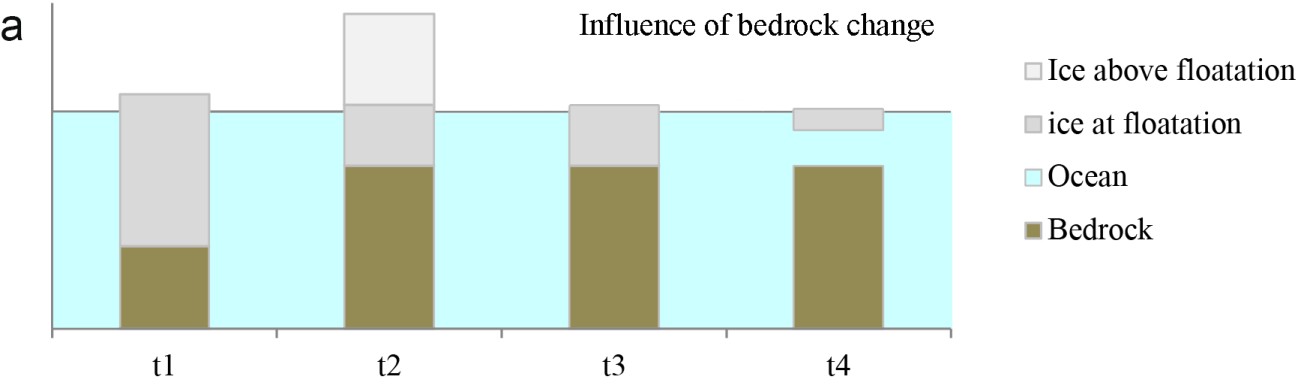

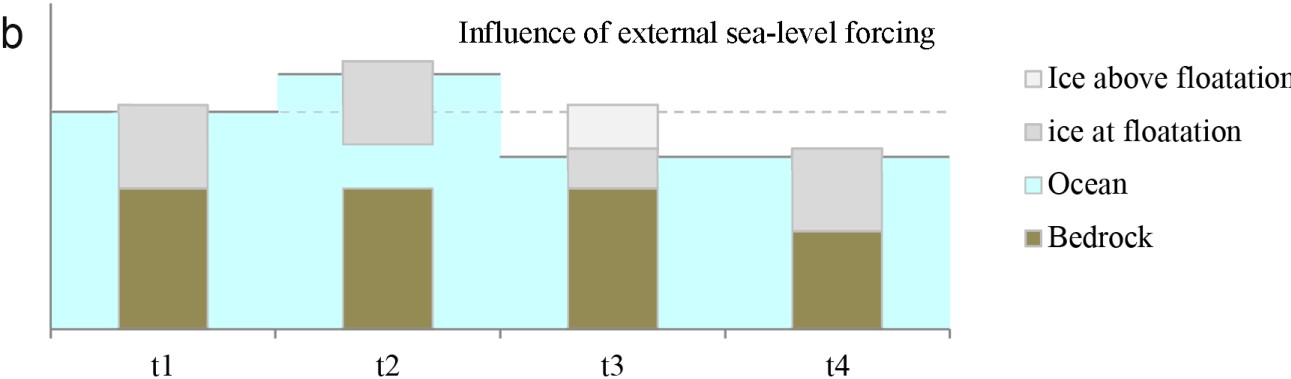

**Figure 2: Geometric evolution of a grid box in time a) including bedrock changes and b) including externally forced sea-level variations.**

The differences in sea-level contribution from $t_1$ to $t_4$ must be independent from the interpretation of what happened between $t_1$ and $t_4$. Hence, bedrock changes have to be taken into account below the ice and in proximity to the ice sheet. The way
5    bedrock changes impact sea level is through changes in the volume of the ocean basins. That is, as bedrock is uplifted, ocean basin volume decreases, leading to a positive sea-level contribution and vice versa.

## 3.    Correcting for bedrock changes

Based on the discussion in the previous section, here we propose an approach to correct the sea-level estimate for bedrock changes. Under floating ice and ice-free ocean, rising bedrock displaces ocean water, and directly leads to a sea-level rise
10    proportional to the bedrock elevation change. The additional sea-level contribution could be calculated from changes in the volume of the ocean water

$$V_{ocean} = \sum_{n} \left( max(-b_n, 0) - H_n \frac{\rho_{ice}}{\rho_{ocean}} \right) \frac{1}{k_n^2} A_n \,, \tag{5}$$

where the term in brackets is the difference between lower ice boundary and bedrock for grid cells containing floating ice and the ocean depth where no ice is present.

However, while bedrock changes under grounded ice have no impact on the estimated ocean volume, they do modify the amount of $V_{af}$, which requires an additional correction. Consider an ice column near flotation but grounded below sea level at $b_0$, with a height above flotation $h_{af}=0$ (e.g. $t_1$ in Figure 2a). When the bedrock rises by a certain amount $\Delta b$ (e.g. transition $t_1$ to $t_2$ in Figure 2a), the ice is lifted and $h_{af}$ (in meter ice equivalent) increases by

$$\Delta h_{af} = \left( \frac{\rho_{water}}{\rho_{ice}} \right) \Delta b. \tag{6}$$

If the sea-level contribution was only computed from differences in total ice volume above flotation ($SLE_{af}$), this would be incorrectly recorded as a sea-level lowering. Furthermore, if the bedrock was lifted to or above sea-level, the final change in $h_{af}$ would equal the ice thickness and

$$\Delta h_{af} = H = - \left( \frac{\rho_{water}}{\rho_{ice}} \right) b_0, \tag{7}$$

where $b_0$ is the initial bedrock elevation (e.g. at $t_1$ in Figure 2a).

In order to consider corrections for bedrock changes under grounded ice, floating ice and ice-free ocean consistently, we chose to modify the ocean volume estimate to incorporate bedrock changes. Note that we assume in the following that all bedrock adjustment occurs within the ice sheet model domain. We suggest to replace the ocean volume calculation above by an estimate of the *potential* ocean volume ($V_{pov}$), i.e. the volume between bedrock and sea level if all ice was instantaneously removed:

$$V_{pov} = \sum_{n} max(-b_n, 0) \frac{1}{k_n^2} A_n, \tag{8}$$

which requires no distinction anymore between grounded and floating ice. However, we have ignored the density difference between ocean water and freshwater, which we will treat separately below.

To convert a change in potential ocean volume to a sea level contribution, $V_{pov}$ has to be divided by the ocean area of typically 3.625 x $10^{14}$ m$^2$:

$$SLC_{pov} = - \left[ \frac{V_{pov}}{A_{ocean}} - \left( \frac{V_{pov}}{A_{ocean}} \right)^{ref} \right]. \tag{9}$$

## 4. Density correction

In this section we discuss the correction necessary to deal with the small difference between fresh water (melted ice) and saline ocean water densities. Transitions of ice below and above flotation and the associated sea-level change can occur both

due to ice mass changes and due to bedrock changes, processes associated with a different density ($\rho_{water}$ vs $\rho_{ocean}$). While changes of $V_{af}$ due to bedrock adjustment and cavity changes are recorded in ocean water equivalent, we must assume that changes in ice sheet mass ultimately contributes to the ocean with a density of fresh water ($\rho_{water}$ = 1000 kg m$^{-3}$). So far, we have calculated all changes in ocean water column equivalent, so now we will apply a density correction for all changes in

ice thickness (above and below flotation).

$$V_{den} = \sum_n H_n \left( \frac{\rho_{ice}}{\rho_{water}} - \frac{\rho_{ice}}{\rho_{ocean}} \right) \frac{1}{k_n^2} A_n \tag{10}$$

and

$$SLC_{den} = - \left[ \frac{V_{den}}{A_{ocean}} - \left( \frac{V_{den}}{A_{ocean}} \right)^{ref} \right] \tag{11}$$

The density ratio $\rho_{water}/\rho_{ocean}$ implies that the correction amounts to ~3 % of the ice volume grounded at/below sea-level.

Finally, to calculate changes in global mean sea-level due to ice-sheet changes, contributions from ice volume above

flotation, potential ocean volume and density correction are added:

$$SLC_{corr} = SLC_{af} + SLC_{pov} + SLC_{den}. \tag{12}$$

## 5.   Externally forced sea-level variations

For long-term ice-sheet simulations, it is common to force ice-sheet models with prescribed variations in (global) sea-level, e.g. representing changes in the northern hemisphere ice sheets when solely simulating the Antarctic ice sheet. For a glacial-interglacial transition the external sea-level forcing (ESLF) may have an amplitude of more than 100 meters and can drive

transitions between floating and grounded ice in the model. In the framework of such simulations, the calculation of sea-level contributions from the ice sheet must be re-considered, because changes in ESLF imply changes in $V_{af}$ of the modelled ice sheet.

We illustrate the implied changes again with a schematic view of one ice column changing over time (Figure 2b). From $t_1$ to $t_2$, the sea-level (horizontal solid line) is increased with respect to the starting value (horizontal dashed line) at constant

bedrock elevation and ice thickness. Consequently, the geometry in the model column changes from just grounded to floating ice (with no sea-level contribution from the ice sheet itself). From $t_2$ to $t_3$ the sea-level is lowered, such that some ice that was floating in $t_2$ is transformed into ice above flotation. At $t_4$, now with combined bedrock change and sea-level change of the same magnitude relative to $t_1$, the ice is just grounded on the lowered bedrock. Calculating the sea-level contribution as described above in Eq. (12), would indicate a change of the contribution from $t_1$ to $t_2$ and $t_3$. However, since these changes

in SL are externally forced, they should not directly contribute to the calculated ice-sheet sea-level contribution itself. For example, the additional volume under the floating ice at $t_2$ occurs because the ice is lifted by the additional, externally-forced

seawater. Equally, the additional ice above flotation created in $t_3$ is merely a consequence of the lower sea-level. Hence, $V_{af}$ has to be corrected to calculate SLC in this case.

This problem can be resolved by calculating changes in $V_{af}$ and $V_{pov}$ for the constructed case where sea-level is fixed and ESLF has no direct impact on the results. Practically, Eqs. (1) and (8) can be modified to compensate changes in $b_n$ that occur solely due to ESLF by corresponding changes in an arbitrary reference level $z_0$, e.g. taken as present-day sea-level, that is time-constant in the absolute reference frame but changes with ESLF (Eqs (13),(14)). In other words, the term $(b_n - z_0)$ is constant with respect to changes in ESLF.

$$V_{af}^0 = \sum_n \left( H_n + min(b_n - z_0, 0) \frac{\rho_{ocean}}{\rho_{ice}} \right) \frac{1}{k_n^2} A_n \tag{13}$$

and

$$V_{pov}^0 = \sum_n max(z_0 - b_n, 0) \frac{1}{k_n^2} A_n. \tag{14}$$

The density correction in Eq. (10) remains unchanged leading with Eqs. (3) and (9) to the corrected sea-level contribution

$$SLC_{corr}^0 = SLC_{af}^0 + SLC_{pov}^0 + SLC_{den}. \tag{15}$$

With this approach, ESLF can be applied for its effect on the flotation condition in the ice sheet model without contaminating the calculation of the sea-level contribution. Note that Equations 13-15 also hold for the case where ESLF is not a spatially uniform value.

## 6. Ice-sheet modelling example

Figure 3 illustrates differences in estimated sea-level contributions for an Antarctic ice-sheet simulation with a model that includes a simplified GIA component and external sea-level forcing (Pattyn, 2017). We have first applied a typical glacial-interglacial experiment (e.g. Golledge et al., 2014; Pollard et al., 2016; Albrecht et al., 2019) over the last 120 kyr (Figure 3a) with the prescribed external sea-level change (based on sea-level reconstructions by Bintanja et al. (2008) and Lambeck et al. (2014)) as a dominant forcing. Atmospheric forcing is produced by perturbing present-day surface temperatures (RACMO2, Van Wessem et al., 2014) with a spatially constant temperature anomaly following ice-core reconstructions from EPICA Dome C (Jouzel et al., 2007), while correcting surface temperatures for elevation changes (e.g. Huybrechts et al., 2002). The second part of the experiment (Figure 3b) continues from the present-day configuration and shows the response to an extreme basal melt forcing applied under floating ice shelves. In this schematic forcing scenario, present-day melt rates are multiplied by a constant factor of 200, resulting in melt rates of up to 100 m yr$^{-1}$ in the Weddell and Ross sea sectors. This extreme melt forcing is not meant to represent a plausible scenario, it only serves to simulate a rapid removal of all floating ice shelves, leading to a retreat of the ice sheet (Pattyn, 2017; Nowicki et al., 2013).

Various SLC corrections and estimates are calculated against the initial configuration in Figure 3a (120 kyr BP) and against the present day configuration in Figure 3b,c. The sea-level contribution calculated from changes in ice volume above flotation ($SLC_{af}$) includes signatures of bedrock and (in the past) externally-forced sea-level changes. In the future retreat scenario (Figure 3b), $SLC_{af}$ is too low compared to our corrected estimate ($SLC_{corr}^0$) mainly because ice volume above

flotation is 'created' by bedrock uplift. This effect of isostatic adjustment on $SLC_{af}$ is exemplified by the steadily decreasing $SLC_{af}$ towards the end of the experiment, while $SLC_{corr}^0$ remains near constant (due to compensating $SLC_{pov}$). Accounting for density differences between ocean and fresh water ($SLC_{den}$) corrects an additional, but smaller underestimation of $SLC_{af}$. The proposed method ($SLC_{corr}^0$) is identical to ($SLC_{corr}$) for the future period (Figure 3b), where no external sea-level forcing is applied, and results in an estimate of the sea-level contribution well above $SLC_{af}$.

In the paleo simulation (Figure 3a), $SLC_{af}$ is biased both by bedrock changes and external sea-level changes. Since $SLC_{pov}$ is calculated in a fixed domain that includes grounded and floating ice and ice-free ocean areas, it is influenced by ice and ocean water loading. In a glaciation scenario with a growing (Antarctic) ice load and decreasing global sea level (Figure 3a, before 15 kyr BP), the correction $SLC_{pov}$ is a combination of a subsiding bedrock under the ice sheet (negative $SLC_{pov}$) and a rising ocean floor in response to reduced water loading (positive $SLC_{pov}$). We remind that the global ocean area A$_{ocean}$ is

assumed as constant here. Although not fully separable, we have estimated the contribution of the two effects by calculating $SLC_{pov}$ within and outside of the glacial ice mask (see supplementary Figure S1). Both effects are of similar magnitude in our setup but $SLC_{pov}$ is slightly dominated by the changing ocean floor outside of the ice mask after periods of rapid sea-level forcing change. In addition, during ice-sheet growth, the negative sea-level excursion in $SLC_{af}$ is exaggerated with increasing amplitude of the external sea-level forcing (cf. $SLC_{af}$ $and$ $SLC_{af}^0$). The proposed method ($SLC_{corr}^0$) results in an

estimate of the negative sea-level contribution in the past of smaller amplitude compared to $SLC_{af}$ and shows that the magnitude and notably the timing of the Last Glacial Maximum low stand are subject to considerable biases in $SLC_{af}$ (Figure 3c). The relative bias in $SLC_{af}$ is larger for stronger ice-sheet retreat (not shown). Accounting for all grounded ice ($SLC_{gr}$) would lead in all cases to the largest excursions in negative and positive sea-level contribution, due to ice grounded below the water level that should mostly be replaced by sea-water. Differences between the different approaches to calculate

SLC become important after 2-3 kyr, roughly corresponding to the shortest response time of bedrock adjustment in the model.

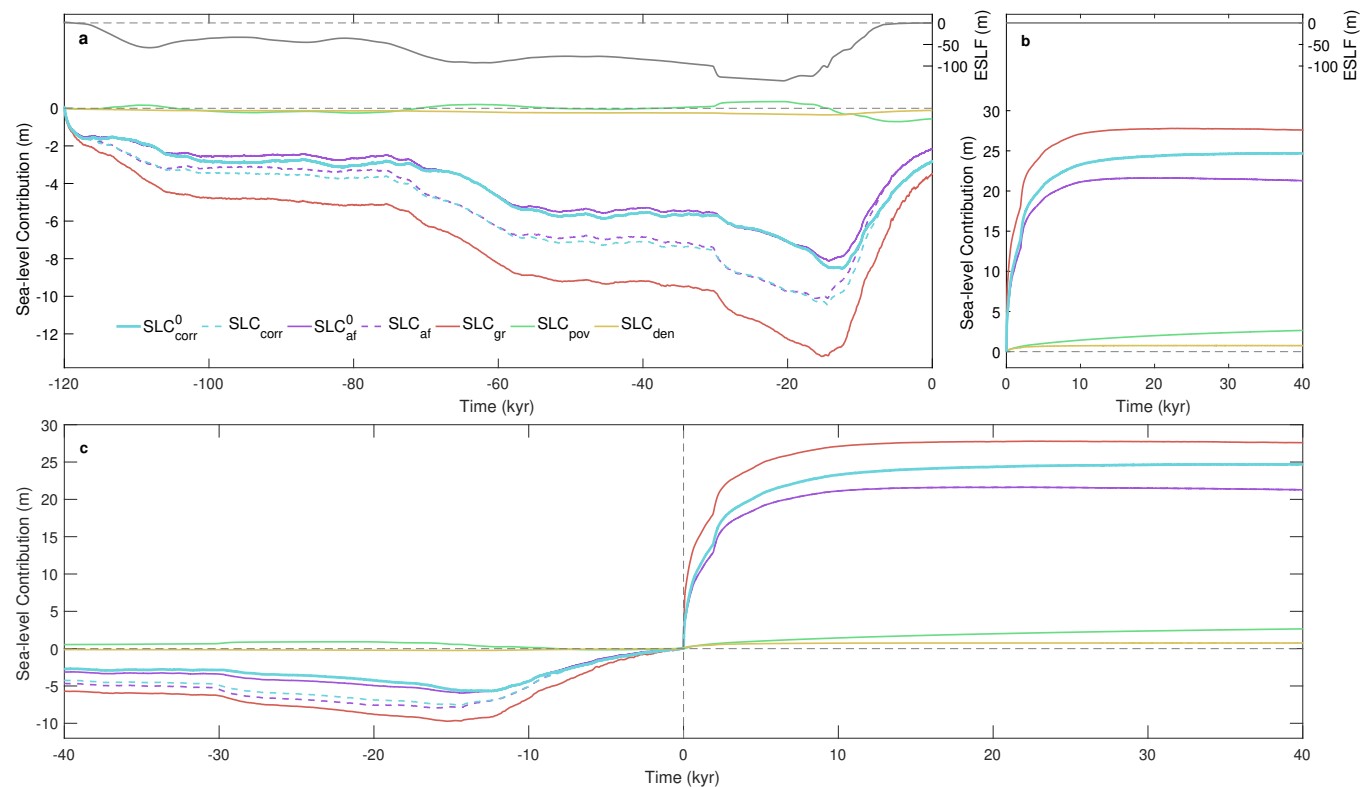

**Figure 3 Different estimates of the sea-level contribution (SLC) from an Antarctic ice-sheet model simulation. (a) Sea-level contribution for the last glacial cycle under external sea-level forcing (ESLF). (b) Schematic deglaciation experiment over the next 40 kyr in which an extreme sub-shelf basal melt perturbation is applied. The model experiment is continuous across year zero, but estimates in (a) and (b) are referenced to the beginning of each period. (c) Same as (a) and (b) combined, but both experiments are referenced to the present-day configuration. Some lines overlap in (b) and for the future in (c) because ESLF is assumed zero for that period. Final corrected sea-level contribution ($SLC_{corr}^0$) calculated at constant external reference sea-level, based on volume above flotation ($SLC_{af}^0$), but corrected for potential ocean volume changes ($SLC_{pov}$) and density ($SLC_{den}$). The dashed lines ($SLC_{corr}$ and $SLC_{af}$) show results calculated for a variable ESLF (grey lines and left y-axis in (a) and (b)) and $SLC_{gr}$ is the sea-level contribution when considering all grounded ice without corrections.**

## 7.  Discussion and conclusions

We have presented a unified approach to calculate the sea-level contribution from a marine ice sheet simulated by an ice-sheet model. The formulation notably corrects for changes in ice volume above flotation in the presence of bedrock changes and external sea-level forcing. In this unified approach, sea-level contributions arise from changes in the ice volume above flotation and potential ocean volume, while changes in external sea-level forcing are corrected for.

When bedrock changes in response to ice loading changes occur under ice that is grounded (below sea-level), changes in potential ocean volume compensate for changes in ice volume above flotation, resulting in a near zero net sea-level contribution as should be expected. Under floating ice (or open ocean), changes in volume above flotation are always zero, but bedrock changes imply ocean depth changes that lead to differences in the sea-level contribution (i.e., due to changes in

ocean basin volume). The combination of changes in ice volume above flotation and potential ocean volume leads to a generalised formulation that is consistent across changes from floating to grounded ice and vice versa.

The region over which ice thickness changes and potential ocean volume changes are calculated must be fixed in time for the comparison and may contain the entire model grid (as done here) or a reasonable subset. It should include all locations that potentially see ice thickness and/or bedrock changes during a simulation. For models with local isostatic adjustment, the region could be the glacial ice mask for paleo simulations and the observed present-day sheet-shelf mask for future simulations dominated by retreat. For non-local isostatic models, the footprint would have to be extended.

In all calculations we have ignored any effects that arise e.g. from water storage in lakes on land and we also did not consider the equation of state of seawater, which implies a non-linear dependence of density on salinity and temperature.

**Data availability.** The SLC time series in Figure 3 and S1 are available as supplement to this publication.

**Author contribution.** HG conceived the project and developed the SLC corrections with assistance of VC. VC performed and analysed the model experiments. HG wrote the manuscript with assistance of all authors.

**Competing interests.** The authors declare that they have no conflict of interest.

**Acknowledgements.** Heiko Goelzer has received funding from the program of the Netherlands Earth System Science Centre (NESSC), financially supported by the Dutch Ministry of Education, Culture and Science (OCW) under Grantnr. 024.002.001. Bas de Boer is funded by the SCOR Corporate foundation for Science. We would like to thank the two reviewers Stephen Price and Rupert Gladstone for their constructive suggestions that helped to improve the manuscript and the editor Ben Galton-Fenzi for guiding the publication process.

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
