# Peer review of "Brief communication: On calculating the sea-level contribution in marine ice-sheet models"

_The Cryosphere, 2019_

## Referee Comment (RC1) · Stephen Price (Referee) · 10 Oct 2019

Summary

This paper discusses the complexities around making estimates of sea-level change (SLC) from models of marine-based ice sheets (i.e., ice sheets with portions of their domains grounded below sea level at some point during the course of a simulation). After pointing out complications that are commonly not addressed or discussed in assessing SLC from marine ice sheet models, the authors propose a mathematical framework for ensuring that such calculations are made consistently. Overall, this is a welcome contribution and one that should be of broad interest to anyone involved in ice sheet modeling for the purposes of estimating SLC. Questions, criticisms, and comments below are mainly in the interest of making the paper more legible and understandable, and thus (hopefully) achieving the authors' goals of having other modelers adopt and use the proposed framework.

Major Comments

It is probably important to note that, while your framework allows for ice sheet models that are coupled to a GIA model, what you consider for the latter here is actually a somewhat limited version of a GIA models. That is, here it is assumed to account strictly for uplift or sinking of the bedrock beneath or proximal to an ice sheet, but does not include other (global) effects, such as SLC due to changes in Earth's rotation, regional SLC due to changes in the Earth's gravitational field, etc. Since some coupled "ice sheet and GIA models" (e.g., Gomez et al.) do account for all of these effects in a consistent way, it might be good to clarify that, here, you separate some of these effects out into "external" sea level forcing.

On that note, the use of "external" to describe all of the "other" ways in which sea level might change could be expanded on a bit (i.e., to clarify what you are lumping in as "external" here). To an ice sheet modeler, this partitioning might seem natural, but to someone else, it might not be immediately clear why you break things up this way.

For Figures 1 and 2, you might consider adding a relative sea-level plot going from one subplot to the next. Specifically, showing the relative change in sea level that occurs as we move from one subpanel to the next. For example, in the top panel of Fig. 1, SLC should be zero, which could be represented by a horizontal line. For the bottom panel of Fig. 1, there would be SLC (a relative increase for both cases I believe?) due to the illustrated changes (increases) in bedrock elevation (which, as shown, imply a decrease in the volume of the ocean basin).

A similar comment applies to Figure 3, but in this case, it would be helpful to include the time series of the "external" sea level forcing.
**TCD**

For the section on including the effects of "external" sea level forcing in your SLC calculation, it would be good to note if Equations 13-14 can be easily adjusted for the case where external sea level forcing is not a uniform value. For example, if external forcing and/or the reference sea level is spatially variable (a function of x, y or lat, lon), can the "$z\_0$" term simply be adjusted to be "$z\_0(i,j)$", where the indices refer to the local (x,y) value of sea level for that particular grid cell? This would be important for the framework to still be useful in the case where external sea level forcing, or the reference sea level, are supplied by a more complex model (i.e., one in which sea level is allowed to vary spatially, as it does in reality).

In general, I think it would help the paper quite a bit if, with each new section where you introduce a new set of terms or corrections, you first state in brief and plain English what that term or correction is and how it affects sea level (some specific examples of this are called out below). While the equations are all carefully laid out and discussed, it seems like you are assuming the reader will naturally and easily parse their importance and meaning, their relative impact on the SLC calculations, etc. It's probably safer to assume that the reader is a bit lazy and help them along right from the start.

When you discuss the impact of changes in bedrock elevation on SLC, in the absence of changes in ice sheet volume, I think it would help to be explicit that the way this impacts sea level is through changes in the volume of the ocean basins. That is, as bedrock is uplifted, ocean basin volume decreases (positive SLC) and as bedrock is lowered, ocean basin volume decreases (negative SLC).

Minor Comments

Minor comments are given in the context of page and line number, e.g. "5,4-10" refers to page 5, lines 4-10.

1,13-14: Make it clear that by "external" forcing of sea level, you mean sea level change that is NOT a result of mass changes of the ice sheet you are considering here.

[Figure]

1,23: " . . . one prognostic variable (ice thickness) . . ." -> " . . . one prognostic variable, ice thickness, . . ."

1,24-25: Again, clarify further what you mean by "external" – NOT from the ice seet being considered here.

1,26: The location of the reference to the Gomez and de Boer models here is odd, and makes it read as if these models are NOT coupled to the sea level equation, when if fact they are some of the few models that ARE. This could be corrected by re-writing the sentence more clearly.

1,26-27: "Consequently, the problem at hand is . . .". This would seem to need some additional wording to be clear. Specifically, here you are assuming that you have an ice sheet model and some form of a GIA model, but that you don't have a fully coupled ice sheet, sea-level model.

2,4-5: Last sentence -> "Our aim here is to provide guidelines and a central reference for . . ."

2,8: "external sea-level changes" – Maybe this is a good place to be a bit more explicit about what you mean by this? Probably the best way to do that is to be very explicit about what you do NOT consider to be external.

2, 14: Clarify if / that you are assuming that bedrock elevation is a negative number if below sea level.

2,16-17: Because you so often refer to what happens in a single column below here, I think you may want to call that out a bit more explicitly when you first introduce it. E.g., something like, "Below, we will often simplify the discussion in order to examine the interplay between ice sheet thickness, bedrock elevation, and sea level for a single column, which can be conceptualized as the values occurring in any single model grid cell (in map view)."

2,24: ". . . but on longer timescales this is not necessarily correct." This idea is left hanging here. It sounds like you intend to note that "A_ocean" can and should be allowed to change over time (at least for the two reference time periods you are calculating SLC over), and that will affect your SLC calculation. But you don't really follow through on that discussion, so it's left a bit ambiguous.

2, Equation 3: I puzzled over the negative sign out in front of the parentheses for a while before I was sure it was correct. You could help the reader here by pointing out that this is necessary since your SLE change is a function of VAF, and a positive change in VAF (increase) over time is associated with a drop in sea level.

3,1-4: It's not clear to me why you discuss the SLC assuming all grounded ice contributes to sea level (here, and elsewhere). No one does this as far as I can tell, so it seems like a weird reference case (if that's what you intend it to be).

3,6-12: You may want to remind the reader here that you are only considering a single column, not the entire ice sheet domain.

3,6-12: Somewhere here, you may want to just be very explicit about how uplift / lowering of bedrock affects relative sea level for an ocean basin of fixed area, and in the absence of any other forcing (i.e., bedrock uplift would be seen as a relative sea level rise and bedrock lowering would be seen as a relative sea level fall).

5,1: "We argue that the differences . . ." -> "The differences . . ."

5,7-9: Here you are explicit about what the impacts of rising / falling bedrock are, but it would be helpful to say this earlier, as you've already gone through Fig. 1 at this point, where this concept is necessary to understand to interpret the figure and discussion.

5,8: "The additional contribution could be calculated from . . ." The use of "could" is confusing as it sounds like you are going to suggest doing something else. If this is what you want us to understand / do (for now), then change "could" to "is".

7,1-7: This section was a little bit opaque to me. I think what you are doing is just coming up with the correction necessary to deal with the small difference between

freshwater (melted snow / ice) and saline ocean water densities. If that is indeed the case, it would help to just come right out and say it explicitly and up front.

7,9-11: I suggest omitting the lead in to this section related to paleo simulations. Since everything you write here applies at any time that sea level is changing – as it always is – I think it would read better to simply start this section as, "External sea-level forcing can drive transitions between floating and grounded ice in the model . . ."

7,16-17: ". . . from just grounded to floating ice [ADD] (with no SLC from the ice sheet itself)."

8, section 6: You should be explicit here up front that you are (presumably?) using an ice sheet model that is coupled to / with a GIA model.

8,19-20: You have multiple "present day[s]" here. Suggesting rewording this as, "Various sea level corrections . . . against the initial configuration at 120 Ka BP (Figure 3a) and the present day configuration (Figure 3b,c)." Noting again that I'm not clear on the two "present days" discussed.

8,20: Again, the use of a scenario where any / all ice grounded below sea level is counted for in the SLC calculation seems odd to me, as no one actually does this. It seems like the base / reference case should be what everyone already does, which is just naively calculate the change in volume above floatation without any of the other corrections you discuss here.

9,5-9: Clarify if you allow the area of the ocean basins to change in these calculations. Is that one of the effects we are seeing here (even if it is buried in the overall change in ocean volume).

9,11: "dominated by the changing ocean floor" – for us to really understand this, I think you need to be clear about whether or not you are talking about open ocean (no ice) or under ice shelves / sheets.

10,15: ". . . while changes in external sea-level forcing are accounted for." It's not clear

what "accounted for" means here. I think what you mean is that you remove or correct for them so that you end up with the SLC contribution from the ice sheet and bedrock changes alone.

10,16: "When bedrock changes occur under ice that is grounded ..." – Do you mean bedrock changes that are independently driven (e.g., by tectonics) or driven in response to changes in the overlying ice volume (GIA)?

11,2: "... that lead to differences in the sea-level contribution [ADD] (i.e., due to changes in ocean basin volume)."

11,3: "... yields unbiased results." Be more specific. This is ambiguous.

Editorial

I have a fairly extensive list of minor, editorial level suggestions that I did not include here, but that can be made available upon request (e.g., through an edited pdf file).
* * *

---

## Referee Comment (RC2) · Rupert Gladstone (Referee) · 18 Oct 2019

The paper aims to bring consistency to the approach (and potentially terminology) used in estimating sea level contributions (SLC) from model-based studies of ice sheets. In general it is successful in this, though I would like to see a clearly defined proposal for appropriate terminology. I think this is a useful contribution to the conversation on SLC due to ice sheets, and would recommend it to be published after some improvements.

The paper is in general fairly clear, but would benefit from a sentence or two at the start of each section providing context and motivation for the direction taken. The detail needs to be broken up with occasional text to orient the reader. I am not altogether sure that this article is short enough to be a "Brief communication", but that is for the

editor to decide!

As to the problem of computing SLC from an ice sheet model, it is not clear whether the paper is trying to say "here is the right way to do it, everyone should do it this way" or "here is one way to do it, think carefully about these issues and use consistent terminology when describing your approach". I would argue it should be the latter, as I think alternative approaches to some aspects of the SLC calculation could be justified. In any case, please try to make this a bit clearer, and if it is the former, the authors need to make stronger arguments (mainly regarding external sea level forcing) why this is the right way!

The paper recommends using the initial sea level as a reference level for calculating SLC. This is a reasonable recommendation, but no consideration is given to other options, such as choosing a different reference level (final sea level? time mean sea level?), or calculating SLC at each timestep based on current sea level (the details of this would probably depend on the numerical scheme being used). Can the authors clarify why their suggestion is the best one? Alternatively, can the authors acknowledge that this is one of several possible approaches that may also be valid? The whole paper is about using ice sheet model outputs to estimate a contribution to mean sea level. It would be good to see a paragraph that puts this in the context of the more complex real world picture. Specifically, there is the gravitational effect of redistribution of ice mass on sea levels. This probably causes a local decrease in sea level, and a far field increase. Also, coupled ice sheet – ocean (possibly also atmosphere) models might be able to simulate a spatial distribution of sea level change, in which it may not be trivial to distinguish between the ice sheet contribution and other effects, and the delivered product could be argued to be "superior" to global mean SLC predictions. Even in such a case, where the modelling approach allows a more complete prediction than simply mean sea level, it may be of value to calculate a mean sea level contribution in order to compare with other ice sheet models, and as such this paper can contribute to this situation also.

Specific comments:

Line 12. I suggest "differences" -> "change" because differences implies comparing two different properties but her I think you mean change over time. Equation 1. The way dx, dy and I,j are used seems to imply (thought this isn't stated) summation over grid cells for a structured rectangular grid. Often in ice sheet modelling unstructured meshes are used, usually either 2D or extruded in the vertical. It should be easy to generalise the notation to all cases except for fully unstructured 3D meshes. Please also clarify in the text that this is an operation over the model grid/mesh.

Equation 4. I don't see what this adds, I would leave it out. If you include it, you need to introduce $V\_gr$.

Page 3.

Line 11. "Ice at floatation" is an odd expression. When there is "ice above floatation" then the "ice at floatation" is clearly grounded ice. So the expression is not intuitive. When we talk about "ice above floatation" we really mean something like "ice above a fictional surface at which, if it formed the actual upper surface, the ice column would be at floatation". The fact is that all this ice is grounded, so talking "ice above floatation" and "ice at floatation" can be a bit confusing. Conceptually, we really mean "ice that can contribute to sea level" and "ice that can't contribute to sea level". I think I would find this discussion generally more …er … "natural" if we talked about a floatation thickness, and so the actual thickness can be above of below this floatation thickness. And ice above the floatation thickness can contribute to sea level whereas ice below cannot. And so bedrock uplift, which doesn't directly impact on ice thickness, does directly impact on the floatation thickness. I am not going to insist on this because the choice of terminology is subjective. But part of the purpose of this paper is to present definitions and terminology with which to discuss ice sheet contribution to sea level rise, so I think some careful thought should be given to this, and my first reaction to "ice at floatation" was that it is somewhat non-intuitive.

Line 12. Perhaps would benefit from a summary sentence ending this paragraph to clarify that an uplift in bedrock will in general lead to SLC being underestimated if only the initial bedrock is used for calculating SLC.

Page 5.

Equation 5. Note that this only calculates ocean volume in the domain of the ice sheet model, which may be a limitation depending on what you want to do with it. I'll read on and find out. . .

Page 6.

This seems a bit clunky. Equations 6 and 7 are only valid in certain situation depending on where the bedrock was or where it is going to. . . can you not go straight to one more generic equation, albeit slightly more complicated, that captures the change in h_af in general, as a function of change in bedrock (and perhaps also of change in thickness)? I am talking here about change just due to the direct bedrock effects, ignoring ocean volume change for the moment.

Line 17. What is a "sea level component"? Perhaps you mean something like "to convert a change in potential ocean volume to a sea level contribution"?

Lines 12-18. It seems that the plan here is to calculate changes in pov? So this would be a purely bedrock contribution, separate from an ice dynamic component.

Equation 9. Right, so now I can see that the limitation of calculating this on the ice model domain is simply that we assume all bedrock adjustment occurs within the ice model domain. This is probably a reasonable and practical assumption to make, but you should clarify that this assumption is being made, with the implication that any study attempting to include bedrock adjustment in an ice sheet model study aimed at projecting sea level rise should endeavour to consider both the extent of the ice sheet itself and of the region over which bedrock adjustment could occur when defining their model domain.

Lines 22-23. "changes in ice mass occur in reality with a density of freshwater". This is a strange sentence. Changes initially occur either at the density of ice (calving) or from ice density to fresh water density (melting), so it is not really correct to say that changes occur "with a density of freshwater". You might want to say something like "Ice loss from the ice sheet ultimately contributes to the ocean with a density of fresh water".

I would also definitely leave out precipitation because this will usually be in the form of snow, initially with a much lower density than ice, and I don't (yet) see why you would need to talk about precipitation?

Page 7.

Equation 10. I don't think $V\_den$ has been clearly defined in the text? Also, I am not clear why this is kept separate from the $V\_af$ calculation. I suppose I must be missing something, but since we've already calculated $V\_af$ in equation 1, why don't we just modify equation 2 like this: $SLE\_af = V\_af/A\_ocn * rho\_ice/rho\_freshwater$ So now we calculate the volume of freshwater being added to the ocean. This is much simpler than what is suggested in the paper. . . why is this wrong? Probably I missed something...

Line 17. What is "t2i"? Just t2 I guess?

Page 8.

Equation 15. This is not "objectively correct", but rather is one way of making the calculation. If externally forced sea level at the end of a simulation is not the same as it was at the start, then should the simulated ice sheet contribution to sea level take into account this change or not? It seems to me one could make justification for different ways of doing this. I think a more objective approach in this paper might be to define terminology for different ways of making this calculation rather than to prescribe what appears to be presented as the "correct" way to do it.

Figure 3. The text is a bit on the small side, and I have to zoom in a lot to see the

subscripts in the legend. Can this be made bigger?

---

## Author Comment (AC1) · 22 Nov 2019

We would like to thank the reviewers for their very constructive comments that helped to improve the manuscript 'Brief communication: On calculating the sea-level contribution in marine ice-sheet models'. We have revised the manuscript accordingly and would be happy to provide a new version

Please find below the reviewer's comments in regular italic and a point-by-point response in bold font.

*Referee #1 (Stephen Price)*

*Summary*
*This paper discusses the complexities around making estimates of sea-level change (SLC) from models of marine-based ice sheets (i.e., ice sheets with portions of their domains grounded below sea level at some point during the course of a simulation). After pointing out complications that are commonly not addressed or discussed in assessing SLC from marine ice sheet models, the authors propose a mathematical framework for ensuring that such calculations are made consistently. Overall, this is a welcome contribution and one that should be of broad interest to anyone involved in ice sheet modeling for the purposes of estimating SLC. Questions, criticisms, and comments below are mainly in the interest of making the paper more legible and understandable, and thus (hopefully) achieving the authors' goals of having other modelers adopt and use the proposed framework.*

**Thank you very much for the positive evaluation and your comments.**

*Major Comments*
*It is probably important to note that, while your framework allows for ice sheet models that are coupled to a GIA model, what you consider for the latter here is actually a somewhat limited version of a GIA models. That is, here it is assumed to account strictly for uplift or sinking of the bedrock beneath or proximal to an ice sheet, but does not include other (global) effects, such as SLC due to changes in Earth's rotation, regional SLC due to changes in the Earth's gravitational field, etc. Since some coupled "ice sheet and GIA models" (e.g., Gomez et al.) do account for all of these effects in a consistent way, it might be good to clarify that, here, you separate some of these effects out into "external" sea level forcing.*

**Agreed. We have added a clarification in the introduction following your suggestion.**

*On that note, the use of "external" to describe all of the "other" ways in which sea level might change could be expanded on a bit (i.e., to clarify what you are lumping in as "external" here). To an ice sheet modeler, this partitioning might seem natural, but to someone else, it might not be immediately clear why you break things up this way.*

**OK. We have added a description in the introduction.**

*For Figures 1 and 2, you might consider adding a relative sea-level plot going from one subplot to the next. Specifically, showing the relative change in sea level that occurs as we move from one subpanel to the next. For example, in the top panel of Fig. 1, SLC should be zero, which could be represented by a horizontal line. For the bottom panel of Fig. 1, there would be SLC (a relative increase for both cases I believe?) due to the illustrated changes (increases) in bedrock elevation (which, as shown, imply a decrease in the volume of the ocean basin).*

**Thanks for the suggestion. We have thought about including SLC information in Figures 1 and 2 before, but decided against it for the following reasons. Until the end of section 2 we use Figures 1 and 2 to illustrate the problem and develop the argument why SLC should be independent of what happens between t1 and t4 (Fig 2a). Showing SLC information in the plot would distract from that line of thought and 'give away' the conclusion. We also believe the figures should be read in a schematic and not quantitative way.**

*A similar comment applies to Figure 3, but in this case, it would be helpful to include the time series of the "external" sea level forcing.*

**Agreed, we have included the external sea-level forcing in Figure 3.**

*For the section on including the effects of "external" sea level forcing in your SLC calculation, it would be good to note if Equations 13-14 can be easily adjusted for the case where external sea level forcing is not a uniform value. For example, if external forcing and/or the reference sea level is spatially variable (a function of x, y or lat, lon), can the "z_0" term simply be adjusted to be "z_0(i,j)", where the indices refer to the local (x,y) value of sea level for that particular grid cell? This would be important for the framework to still be useful in the case where external sea level forcing, or the reference sea level, are supplied by a more complex model (i.e., one in which sea level is allowed to vary spatially, as it does in reality).*

**Yes, agreed. The formulation in Equ. 13-14 remains valid for non-uniform changes of the external sea-level forcing. We have added a clarification in the text to confirm that.**

*In general, I think it would help the paper quite a bit if, with each new section where you introduce a new set of terms or corrections, you first state in brief and plain English what that term or correction is and how it affects sea level (some specific examples of this are called out below). While the equations are all carefully laid out and discussed, it seems like you are assuming the reader will naturally and easily parse their importance and meaning, their relative impact on the SLC calculations, etc. It's probably safer to assume that the reader is a bit lazy and help them along right from the start.*

**Agreed. We have added short introductions to section 3, 4 and 5 accordingly.**

*When you discuss the impact of changes in bedrock elevation on SLC, in the absence of changes in ice sheet volume, I think it would help to be explicit that the way this impacts sea level is through changes in the volume of the ocean basins. That is, as bedrock is uplifted, ocean basin volume decreases (positive SLC) and as bedrock is lowered, ocean basin volume decreases (negative SLC).*

**OK, we have included a description as suggested.**

*Minor Comments*
*Minor comments are given in the context of page and line number, e.g. "5,4-10" refers to page 5, lines 4-10.*

*1,13-14: Make it clear that by "external" forcing of sea level, you mean sea level change that is NOT a result of mass changes of the ice sheet you are considering here.*

**OK, added a sentence in the abstract.**

*1,23: " . . . one prognostic variable (ice thickness) . . ." -> " . . . one prognostic variable, ice thickness, . . ."*

**OK, corrected.**

*1,24-25: Again, clarify further what you mean by "external" – NOT from the ice seet being considered here.*

**OK, added a clarification.**

*1,26: The location of the reference to the Gomez and de Boer models here is odd, and makes it read as if these models are NOT coupled to the sea level equation, when if fact they are some of the few models that ARE. This could be corrected by re-writing the sentence more clearly.*

**Yes, agreed. The sentence has been reformulated.**

*1,26-27: "Consequently, the problem at hand is . . .". This would seem to need some additional wording to be clear. Specifically, here you are assuming that you have an ice sheet model and some form of a GIA model, but that you don't have a fully coupled ice sheet, sea-level model.*

**OK, included in the changes suggested in the major comments.**

*2,4-5: Last sentence -> "Our aim here is to provide guidelines and a central reference for ..."*

**OK, modified as suggested.**

*2,8: "external sea-level changes" – Maybe this is a good place to be a bit more explicit about what you mean by this? Probably the best way to do that is to be very explicit about what you do NOT consider to be external.*

**OK. We have captured that in the extended discussion above.**

*2, 14: Clarify if / that you are assuming that bedrock elevation is a negative number if below sea level.*

**OK, added clarification in the text.**

*2,16-17: Because you so often refer to what happens in a single column below here, I think you may want to call that out a bit more explicitly when you first introduce it. E.g.,*

*something like, "Below, we will often simplify the discussion in order to examine the interplay between ice sheet thickness, bedrock elevation, and sea level for a single column, which can be conceptualized as the values occurring in any single model grid cell (in map view)."*

**Thank you, included as suggested.**

*2,24: ". . . but on longer timescales this is not necessarily correct." This idea is left hanging here. It sounds like you intend to note that "A_ocean" can and should be allowed to change over time (at least for the two reference time periods you are calculating SLC over), and that will affect your SLC calculation. But you don't really follow through on that discussion, so it's left a bit ambiguous.*

**OK, Included a clarification in the text. "Estimating changes in Aocean would require a fully-coupled global GIA-ice sheet-sea-level model."**

*2, Equation 3: I puzzled over the negative sign out in front of the parentheses for a while before I was sure it was correct. You could help the reader here by pointing out that this is necessary since your SLE change is a function of VAF, and a positive change in VAF (increase) over time is associated with a drop in sea level.*

**OK, added a clarification in the text.**

*3,1-4: It's not clear to me why you discuss the SLC assuming all grounded ice contributes to sea level (here, and elsewhere). No one does this as far as I can tell, so it seems like a weird reference case (if that's what you intend it to be).*

**SLC_gr is not used as a reference, but we believe it is still interesting to compare to. Especially since the our final estimate in the future lies between SLC_gr and SLC_af. Therefore, we would like to keep it. We have added a clarification to that end in the manuscript: "… estimating the sea-level contribution instead from the entire grounded ice volume Vgr (Eq. (4)) can lead to considerable biases and is only used for comparison here."**

*3,6-12: You may want to remind the reader here that you are only considering a single column, not the entire ice sheet domain.*

**OK, added clarification in the text.**

*3,6-12: Somewhere here, you may want to just be very explicit about how uplift / lowering of bedrock affects relative sea level for an ocean basin of fixed area, and in the absence of any other forcing (i.e., bedrock uplift would be seen as a relative sea level rise and bedrock lowering would be seen as a relative sea level fall).*

**We have added a clarifying sentence at the end of the section to not destroy the story line we are setting up.**

*5,1: "We argue that the differences . . ." -> "The differences . . ."*

**OK**

*5,7-9: Here you are explicit about what the impacts of rising / falling bedrock are, but it would be helpful to say this earlier, as you've already gone through Fig. 1 at this point, where this concept is necessary to understand to interpret the figure and discussion.*

**OK, this has been added at the end of section 2.**

*5,8: "The additional contribution could be calculated from ..." The use of "could" is confusing as it sounds like you are going to suggest doing something else. If this is what you want us to understand / do (for now), then change "could" to "is".*

**We suggest doing something different below: "We suggest to replace the ocean volume calculation above by an estimate of the potential ocean volume".**

*7,1-7: This section was a little bit opaque to me. I think what you are doing is just coming up with the correction necessary to deal with the small difference between freshwater (melted snow / ice) and saline ocean water densities. If that is indeed the case, it would help to just come right out and say it explicitly and up front.*

**Agreed. We have included an introductory sentence following the suggestion.**

*7,9-11: I suggest omitting the lead in to this section related to paleo simulations. Since everything you write here applies at any time that sea level is changing – as it always is –*

*I think it would read better to simply start this section as, "External sea-level forcing can drive transitions between floating and grounded ice in the model . . ."*

**We have taken away the paleo reference, but have kept the rest of the introduction to make clear what the external forcing represents and what it is typically used for.**

*7,16-17: ". . .from just grounded to floating ice [ADD] (with no SLC from the ice sheet itself)."*

**OK**

*8, section 6: You should be explicit here up front that you are (presumably?) using an ice sheet model that is coupled to / with a GIA model.*

**OK**

*8,19-20: You have multiple "present day[s]" here. Suggesting rewording this as, "Various sea level corrections . . . against the initial configuration at 120 Ka BP (Figure 3a) and the present day configuration (Figure 3b,c)." Noting again that I'm not clear on the two "present days" discussed.*

**OK, reformulated as suggested.**

*8,20: Again, the use of a scenario where any / all ice grounded below sea level is counted for in the SLC calculation seems odd to me, as no one actually does this. It seems like the base / reference case should be what everyone already does, which is just naively calculate the change in volume above floatation without any of the other corrections you discuss here.*

**While we decided to keep SLC_gr in the discussion for comparison (see also response to general point), we have moved it to the end of the section to give it less weight.**

*9,5-9: Clarify if you allow the area of the ocean basins to change in these calculations. Is that one of the effects we are seeing here (even if it is buried in the overall change in ocean volume).*

**OK, Added a clarification in the text: "We remind that the global ocean area Aocean is assumed as constant here."**

*9,11: "dominated by the changing ocean floor" – for us to really understand this, I think you need to be clear about whether or not you are talking about open ocean (no ice) or under ice shelves / sheets.*

**OK, added "outside of the ice mask" to make that clearer.**

*10,15: ". . . while changes in external sea-level forcing are accounted for." It's not clear what "accounted for" means here. I think what you mean is that you remove or correct for them so that you end up with the SLC contribution from the ice sheet and bedrock changes alone.*

**Ok, reformulated.**

*10,16: "When bedrock changes occur under ice that is grounded . . ." – Do you mean bedrock changes that are independently driven (e.g., by tectonics) or driven in response to changes in the overlying ice volume (GIA)?*

**We mean GIA. Added a clarification.**

*11,2: "... that lead to differences in the sea-level contribution [ADD] (i.e., due to changes in ocean basin volume)."*

**OK, added as suggested**

*11,3: ". . . yields unbiased results." Be more specific. This is ambiguous.*

**OK, reformulated to emphasize consistent results for transitions between floating and grounded ice.**

*Editorial*
*I have a fairly extensive list of minor, editorial level suggestions that I did not include here, but that can be made available upon request (e.g., through an edited pdf file).*

**We would be happy to receive the commented pdf file to improve our manuscript further.**

**Thanks again for reviewing this paper.**

*Referee #2 (Rupert Gladstone)*

*The paper aims to bring consistency to the approach (and potentially terminology) used in estimating sea level contributions (SLC) from model-based studies of ice sheets. In general it is successful in this, though I would like to see a clearly defined proposal for appropriate terminology. I think this is a useful contribution to the conversation on SLC due to ice sheets, and would recommend it to be published after some improvements. The paper is in general fairly clear, but would benefit from a sentence or two at the start of each section providing context and motivation for the direction taken. The detail needs to be broken up with occasional text to orient the reader. I am not altogether sure that this article is short enough to be a "Brief communication", but that is for the editor to decide!*

**Thank you very much for the positive evaluation. We have improved the manuscript by reconsidering the terminology and adding introductory sentences in each section. We believe that even after the revisions the formal conditions for a "Brief communication" are met by our manuscript.**

*As to the problem of computing SLC from an ice sheet model, it is not clear whether the paper is trying to say "here is the right way to do it, everyone should do it this way" or "here is one way to do it, think carefully about these issues and use consistent terminology when describing your approach". I would argue it should be the latter, as I think alternative approaches to some aspects of the SLC calculation could be justified. In any case, please try to make this a bit clearer, and if it is the former, the authors need to make stronger arguments (mainly regarding external sea level forcing) why this is the right way!*

**We fully agree with the reviewer on the second mentioned aim of the paper ("here is one way to do it, …") and have clarified that by carefully revising the text in the paper.**

*The paper recommends using the initial sea level as a reference level for calculating SLC. This is a reasonable recommendation, but no consideration is given to other options, such as choosing a different reference level (final sea level? time mean sea level?), or calculating SLC at each timestep based on current sea level (the details of this would probably depend on the numerical scheme being used). Can the authors clarify why their suggestion is the best one? Alternatively, can the authors acknowledge that this is one of several possible approaches that may also be valid?*

**The choice of the reference sea-level is completely arbitrary, so other options are equally possible and correct. We say in the text "The actual sea-level contribution […] is typically calculated relative to a reference value, often the present day (modelled) configuration or the configuration at the start or end of an experiment", which in our reading gives up to three different options and does not prioritise any of them. In Figure 3 we also show results for two different reference levels.**

*The whole paper is about using ice sheet model outputs to estimate a contribution to mean sea level. It would be good to see a paragraph that puts this in the context of the more complex real world picture. Specifically, there is the gravitational effect of redistribution of ice mass on sea levels. This probably causes a local decrease in sea level, and a far field increase. Also, coupled ice sheet – ocean (possibly also atmosphere) models might be able to simulate a spatial distribution of sea level change, in which it may not be trivial to distinguish between the ice sheet contribution and other effects, and the delivered product could be argued to be "superior" to global mean SLC predictions. Even in such a case, where the modelling approach allows a more complete prediction than simply mean sea level, it may be of value to calculate a mean sea level contribution in order to compare with other ice sheet models, and as such this paper can contribute to this situation also.*

**We have expanded the introduction by adding more information about the "full problem" and models that solve the coupled ice sheet-GIA-sea-level problem.**

*Specific comments:*

*[Page 2]*
*Line 12. I suggest "differences" -> "change" because differences implies comparing two different properties but her I think you mean change over time.*

**OK, modified as suggested.**

*Equation 1. The way dx, dy and I,j are used seems to imply (thought this isn't stated) summation over grid cells for a structured rectangular grid. Often in ice sheet modelling unstructured meshes are used, usually either 2D or extruded in the vertical. It should be easy to generalise the notation to all cases except for fully unstructured 3D meshes. Please also clarify in the text that this is an operation over the model grid/mesh.*

**Yes, thank you. We have modified all equations using the more general formulation summing over elements of a grid.**

*Equation 4. I don't see what this adds, I would leave it out. If you include it, you need to introduce V_gr.*

**Please see response to Reviewer 1 above on the same topic. V_gr has been introduced.**

*Page 3.*
*Line 11. "Ice at floatation" is an odd expression. When there is "ice above floatation" then the "ice at floatation" is clearly grounded ice. So the expression is not intuitive. When we talk about "ice above floatation" we really mean something like "ice above a fictional surface at which, if it formed the actual upper surface, the ice column would be at floatation". The fact is that all this ice is grounded, so talking "ice above floatation" and "ice at floatation" can be a bit confusing. Conceptually, we really mean "ice that can contribute to sea level" and "ice that can't contribute to sea level". I think I would find this discussion generally more ...er ... "natural" if we talked about a floatation thickness, and so the actual thickness can be above of below this floatation thickness. And ice above the floatation thickness can contribute to sea level whereas ice below cannot. And so bedrock uplift, which doesn't directly impact on ice thickness, does directly impact on the floatation thickness. I am not going to insist on this because the choice of terminology is subjective. But part of the purpose of this paper is to present definitions and terminology with which to discuss ice sheet contribution to sea level rise, so I think some careful thought should be given to this, and my first reaction to "ice at floatation" was that it is somewhat non-intuitive.*

**We agree that the term "ice at floatation" is uncommon and have removed it from the manuscript. Instead, we introduce and use "floatation thickness" as suggested.**

*Line 12. Perhaps would benefit from a summary sentence ending this paragraph to clarify that an uplift in bedrock will in general lead to SLC being underestimated if only the initial bedrock is used for calculating SLC.*

**We have added a sentence on the role of bedrock changes at the end of the section. Please also compare response to Reviewer 1.**

*Page 5.*
*Equation 5. Note that this only calculates ocean volume in the domain of the ice sheet model, which may be a limitation depending on what you want to do with it. I'll read on and find out. . .*

**Yes, please see comment below.**

*Page 6.*
*This seems a bit clunky. Equations 6 and 7 are only valid in certain situation depending on where the bedrock was or where it is going to. . . can you not go straight to one more generic equation, albeit slightly more complicated, that captures the change in h_af in general, as a function of change in bedrock (and perhaps also of change in thickness)? I am talking here about change just due to the direct bedrock effects, ignoring ocean volume change for the moment.*

**We chose this way of laying out the problem to first make clear why using $V_{af}$ alone is not correct, before going to the generalised solution. We believe this is instructive since using $V_{af}$ alone is what most people are used to.**

*Line 17. What is a "sea level component"? Perhaps you mean something like "to convert a change in potential ocean volume to a sea level contribution"?*

**OK, reformulated.**

*Lines 12-18. It seems that the plan here is to calculate changes in pov? So this would be a purely bedrock contribution, separate from an ice dynamic component.*

**Yes, the calculation is only dependent on bedrock changes.**

*Equation 9. Right, so now I can see that the limitation of calculating this on the ice model domain is simply that we assume all bedrock adjustment occurs within the ice model domain. This is probably a reasonable and practical assumption to make, but you should clarify that this assumption is being made, with the implication that any study attempting to include bedrock adjustment in an ice sheet model study aimed at projecting sea level rise should endeavour to consider both the extent of the ice sheet itself and of the region over which bedrock adjustment could occur when defining their model domain.*

**We have added a clarification in the text: "Note that we assume in the following that all bedrock adjustment occurs within the ice sheet model domain." And we also have a discussion item precisely on that question in the final section of the manuscript.**

*Lines 22-23. "changes in ice mass occur in reality with a density of freshwater". This is a strange sentence. Changes initially occur either at the density of ice (calving) or from ice density to fresh water density (melting), so it is not really correct to say that changes occur "with a density of freshwater". You might want to say something like "Ice loss from the ice sheet ultimately contributes to the ocean with a density of fresh water".*
*I would also definitely leave out precipitation because this will usually be in the form of snow, initially with a much lower density than ice, and I don't (yet) see why you would need to talk about precipitation?*

**OK, reformulated the text accordingly.**

*Page 7.*
*Equation 10. I don't think V_den has been clearly defined in the text? Also, I am not clear why this is kept separate from the V_af calculation. I suppose I must be missing something, but since we've already calculated V_af in equation 1, why don't we just modify equation 2 like this: SLE_af = V_af/A_ocn * rho_ice/rho_freshwater So now we calculate the volume of freshwater being added to the ocean. This is much simpler than what is suggested in the paper. . . why is this wrong? Probably I missed something...*

**V_den is treated separately, because the correction needs to be applied for all ice volume changes, including ice below flotation. This would not be the case if equation 2 was modified as suggested. An additional correction would still be required for the ice below flotation. We believe it is clearer to apply the correction for all ice volume changes.**

**We have added a clarification in the text "… we will apply a density correction for all changes in ice thickness (both above and below flotation)."**

*Line 17. What is "t2i"? Just t2 I guess?*

**Yes, corrected typo.**

*Page 8.*
*Equation 15. This is not "objectively correct", but rather is one way of making the calculation. If externally forced sea level at the end of a simulation is not the same as it was at the start, then should the simulated ice sheet contribution to sea level take into account this change or not? It seems to me one could make justification for different ways of doing this. I think a more objective approach in this paper might be to define terminology for different ways of making this calculation rather than to prescribe what appears to be presented as the "correct" way to do it.*

**We fully agree that there are different and equally valid ways to make the calculation. However, following the argumentation illustrated in Fig 2b, we cannot see any scenario in which contaminating the sea-level estimate with the external sea-level forcing is desirable. And this argument is independent of the choice of reference sea-level.**
**We have resolved what we believe to be a misunderstanding about the role of the reference level z0 by reformulating the description of Equ 13 and 14. The reference level z0 is only needed to compensate for changes in bedrock that are solely based on changes of ESLF. We have also clarified in the text that the choice of z0 is completely arbitrary. To give an example: for a an instantaneous change of ESLF to -100 m, b (measured in the model relative to sea-level) would increase by 100 m, which is compensated in turn by -z0 (equally measured in the model relative to sea-level) .**

*Figure3. The text is a bit on the small side, and I have to zoom in a lot to see the subscripts in the legend. Can this be made bigger?*

**Yes, the figure has been updated with larger text in the legend.**

**Thanks again for a constructive review of our manuscript.**